# CLIC4 and CLIC1 bridge plasma membrane and cortical actin network for a successful cytokinesis

Zeynep Cansu Uretmen Kagiali[1] , Nazan Saner[1] , Mehmet Akdag[1], Erdem Sanal[1], Beste Senem Degirmenci[1], Gurkan Mollaoglu[1], Nurhan Ozlu[1,2]

**CLIC4 and CLIC1 are members of the well-conserved chloride intracellular channel proteins (CLICs) structurally related to glutathione-S-transferases. Here, we report new roles of CLICs in cytokinesis. At the onset of cytokinesis, CLIC4 accumulates at the cleavage furrow and later localizes to the midbody in a RhoA-dependent manner. The cell cycle–dependent localization of CLIC4 is abolished when its glutathione S-transferase activity–related residues (C35A and F37D) are mutated. Ezrin, anillin, and ALIX are identified as interaction partners of CLIC4 at the cleavage furrow and midbody. Strikingly, CLIC4 facilitates the activation of ezrin at the cleavage furrow and reciprocally inhibition of ezrin activation diminishes the translocation of CLIC4 to the cleavage furrow. Furthermore, knockouts of CLIC4 and CLIC1 cause abnormal blebbing at the polar cortex and regression of the cleavage furrow at late cytokinesis leading to multinucleated cells. We conclude that CLIC4 and CLIC1 function together with ezrin where they bridge plasma membrane and actin cytoskeleton at the polar cortex and cleavage furrow to promote cortical stability and successful completion of cytokinesis in mammalian cells.**

## Introduction

The chloride intracellular channel (CLIC) family is composed of six members (CLIC1-6) that can exist in both soluble and membrane-integrated forms. CLICs are highly conserved from invertebrates with homologs in *Drosophila melanogaster* (*Dm*CLIC) and *Caenorhabditis elegans* (EXC4 and EXL1) to mammals suggesting an essential role in metazoans (Singh, 2010). CLIC4 is the most studied member of the CLIC family and has been implicated in many actin-based cellular processes including G-protein–coupled receptor signaling, cell differentiation, adhesion, spreading, migration, integrin signaling, and protein trafficking (Ponsioen et al, 2009; Argenzio et al, 2014, 2018; Shukla et al, 2014; Chou et al, 2016). However, how CLIC4 functions in these diverse cellular processes has not been completely resolved.

Growing evidence highlights the functioning of CLIC4 in an actin-mediated manner. Previously, cytosolic CLIC4 was shown to transiently translocate to the plasma membrane upon serum or lysophosphatidic acid (LPA)–induced RhoA activation in an F-actin–dependent manner (Ponsioen et al, 2009). In concordance with this, CLIC4 was found to directly interact with the G-actin–binding protein profilin-1 and was identified as a component in RhoA-mDia2 signaling that induces cortical actin polymerization (Argenzio et al, 2018). Moreover, CLIC4 regulates the formation of branched actin networks on the early endosomes. Consequently, its depletion leads to massive accumulation of branched actin on the surface of early endosomes, which interferes with cargo transport and vesicular trafficking (Chou et al, 2016). CLIC4 is recruited to $\beta$1-integrin at the plasma membrane upon LPA stimulation, and its knockdown causes a reduced integrin-mediated cell adhesion and increased motility (Argenzio et al, 2014).

Strikingly, multiple studies implicated CLIC4 in cancer progression (Peretti et al, 2015), but the underlying molecular mechanisms remain to be elucidated. CLIC4 expression is reported to be induced by the oncogene c-Myc, tumor necrosis factor TNF-$\alpha$, and tumor suppressor p53 (Fernandez-Salas et al, 1999; Shiio et al, 2006). In addition, in many human epithelial cancers, CLIC4 expression was lost in tumor cells, whereas it was gained in tumor stroma during cancer pathogenesis (Suh et al, 2007). The expression level of CLIC4 was found to be gradually decreased in squamous cancer cells as they transformed from benign to malignant (Suh et al, 2012). For this, investigating the specific function of CLIC4 in cell division would greatly help our understanding of its contribution to carcinogenesis. CLICs have not been examined in detail in the context of cell division, except for early studies implying their involvement in cell cycle regulation (Valenzuela et al, 2000; Berryman & Goldenring, 2003).

The soluble form of CLICs is structurally related to omega-type glutathione-S-transferases (GST-omega) (Dulhunty et al, 2001; Littler et al, 2005; Edwards & Kahl, 2010), which suggests glutathione (GSH)-dependent enzymatic activity for the CLIC family. In consistent with this, CLICs exhibit GSH-dependent oxidoreductase activity in vitro (Al Khamici et al, 2015). Furthermore, CLIC3 has been recently shown to promote the invasive behavior of cancer cells through its GSH-dependent oxidoreductase activity (Hernandez-Fernaud et al, 2017). However, the in vivo enzymatic activity of other members of

[1]Department of Molecular Biology and Genetics, Koç University, Istanbul, Turkey   [2]Koç University Research Center for Translational Medicine (KUTTAM), Istanbul, Turkey

Correspondence: nozlu@ku.edu.tr

CLICs, as well as their substrates and function remain to be discovered.

Our previous proteomics study investigating the biochemical changes at the cell surface during cell division revealed a significant enrichment of both CLIC4 and CLIC1 on the surface of rounded up mitotic cells compared with flat interphase cells (Ozlu et al, 2015). Here, we investigated the dynamics and the role of CLIC4 and CLIC1 during cell division and showed that both are involved in the progression of cytokinesis. CLIC4 localizes to the cleavage furrow and midbody during cytokinesis in a RhoA activation–dependent manner via its conserved residues Cys35 and Phe37 that are critical for the putative substrate binding of CLIC4 (Ponsioen et al, 2009; Argenzio et al, 2018). Comparing the interaction networks of CLIC4 and its mutant (C35A) at cytokinesis allowed us to identify ezrin, anillin, and ALIX as critical partners of CLIC4 during cytokinesis. Strikingly, CLIC4 stimulates the activation of ezrin, a member of the ezrin-radixin–moesin (ERM) family, which reciprocally facilitates the translocation of CLIC4 to the cleavage furrow. Double knockout of CLIC4 and CLIC1 impairs the stability of membrane-cortical actin interaction and leads to abnormal blebbing and abscission defect during cytokinesis. The accumulation of CLIC4 and CLIC1 at blebs suggests their importance in maintaining bleb retraction and cortical stability. We propose that as part of a plasma membrane–actin cytoskeleton anchoring complex, CLIC4 and CLIC1 are essential for mammalian cytokinesis.

# Results

## Spatiotemporal regulation of CLIC4 during cell division is dependent on RhoA activation and requires Cys35 and Phe37 residues

To examine the spatiotemporal regulation of CLIC4 during cell division, CLIC4-GFP expressing HeLa S3 cells were visualized using live-cell imaging. Consistent with our previous proteomics data (Ozlu et al, 2015), CLIC4 decorated the cell surface as the cell rounded up at the metaphase (Fig 1A and Video 1). Strikingly, as the sister chromatids separated at the anaphase onset, CLIC4 started disappearing from the cell poles and gradually accumulating at the cleavage furrow (Fig 1A and Video 1). The immunostaining of endogenous CLIC4 revealed a similar dynamic localization pattern during cell division (Fig S1A).

RhoA activation is one of the key regulatory steps of cytokinesis, which mediates the formation of the contractile ring and cleavage furrow (Piekny et al, 2005). To further assess whether the cleavage furrow localization of CLIC4 is specifically dependent on RhoA activation, HeLa S3 cells were synchronized to cytokinesis and treated with Rhosin, which inhibits the GEF activation domain of RhoA (Shang et al, 2012). Endogenous CLIC4 immunostaining in cytokinesis cells revealed that the inhibition of RhoA activation significantly reduces CLIC4 accumulation at the cleavage furrow (Fig 1B and C).

Previously, six conserved residues of CLIC4, including Cys35 and Phe37, were identified as essential for the translocation of cytosolic CLIC4 to the plasma membrane in response to RhoA activation

(Ponsioen et al, 2009). Both Cys35 and Phe37 residues are structurally critical because their equivalents in GST-omega 1 are located at the enzymatically active cleft and the glutathione-binding site, respectively. To address the importance of Cys35 and Phe37 in RhoA-dependent cleavage furrow localization of CLIC4, we analyzed the subcellular localization of CLIC4-WT-GFP, CLIC4-C35A-GFP, and CLIC4-F37D-GFP during cytokinesis (Fig 1D). The expression levels of CLIC4-GFP proteins were similar to the expression level of endogenous CLIC4 protein (Fig S1B). Only the wild-type CLIC4 accumulated at the cleavage furrow but not the mutant CLIC4 proteins (Fig 1D and E). Live-cell imaging of CLIC4-C35A-GFP and CLIC4-F37D-GFP revealed that not only the cleavage furrow localization but also the mitotic cell surface localization of mutant CLIC4 proteins were altered (Fig 1F and Video 2). Based on these results, we conclude that CLIC4 localizes to the cleavage furrow and the midbody during cytokinesis in response to RhoA activation via its conserved residues Cys35 and Phe37.

## A systematic comparison between proximity interactomes of wild-type and mutant CLIC4 proteins reveals novel interactors of CLIC4 in cytokinesis

To gain insight into the mechanisms behind the RhoA-dependent cleavage furrow localization of CLIC4, we attempted to identify interaction partners of CLIC4 that might be involved in its targeting. For this, we decided to uncover the differences in interaction partners of wild-type and mutant CLIC4 (C35A) during cytokinesis. We focused on the CLIC4-C35A mutant, as active Cys35 was suggested to be critical for the potential substrate-binding of CLIC4 (Ponsioen et al, 2009), and the mutation of this residue abolished the cleavage furrow localization of CLIC4 (Fig 1D–F).

Because of the transient nature and the difficulties in achieving high synchronization efficiency of the cytokinesis phase, full biochemical mapping of the CLIC4 interactome is a challenging undertaking. This prompted us to apply a well-established monopolar cytokinesis arrest. Monopolar cytokinesis has been shown to be a good surrogate for the biochemical analysis of cytokinesis (Hu et al, 2008; Ozlu et al, 2010; Karayel et al, 2018). As monopolar cytokinesis cells possess a broader cleavage furrow than bipolar cells, this provides a wider recruitment area for CLIC4. Similar to bipolar cytokinesis, only the CLIC4-WT-GFP is highly concentrated at the cleavage furrow in monopolar cytokinesis cells, but not the CLIC4-C35A-GFP (Fig 2A). This confirms that monopolar cytokinesis recapitulates the CLIC4 localization pattern of bipolar cytokinesis and provides better synchrony for the following biochemical experiments.

Given the transient and dynamic nature of CLIC4 interactions, traditional immunoprecipitation assays were not efficient in identifying its interacting partners (Argenzio et al, 2014). To overcome this limitation, we took the biotin identification (BioID) proximity labeling approach in live cells (Roux et al, 2012). We generated mycBirA*-CLIC4-WT and mycBirA*-CLIC4-C35A–expressing cells. Only mycBirA*-CLIC4-WT was able to localize to the cleavage furrow in monopolar cytokinesis cells as seen by strong myc-tag immunostaining (Fig 2B). Cells were cultured either with or without supplemental biotin (50 $\mu$M) (Fig 2C, right and left panels, respectively), and affinity purification of biotinylated proteins was performed via streptavidin pull-down. In the absence of supplemental biotin, only the endogenous biotinylated

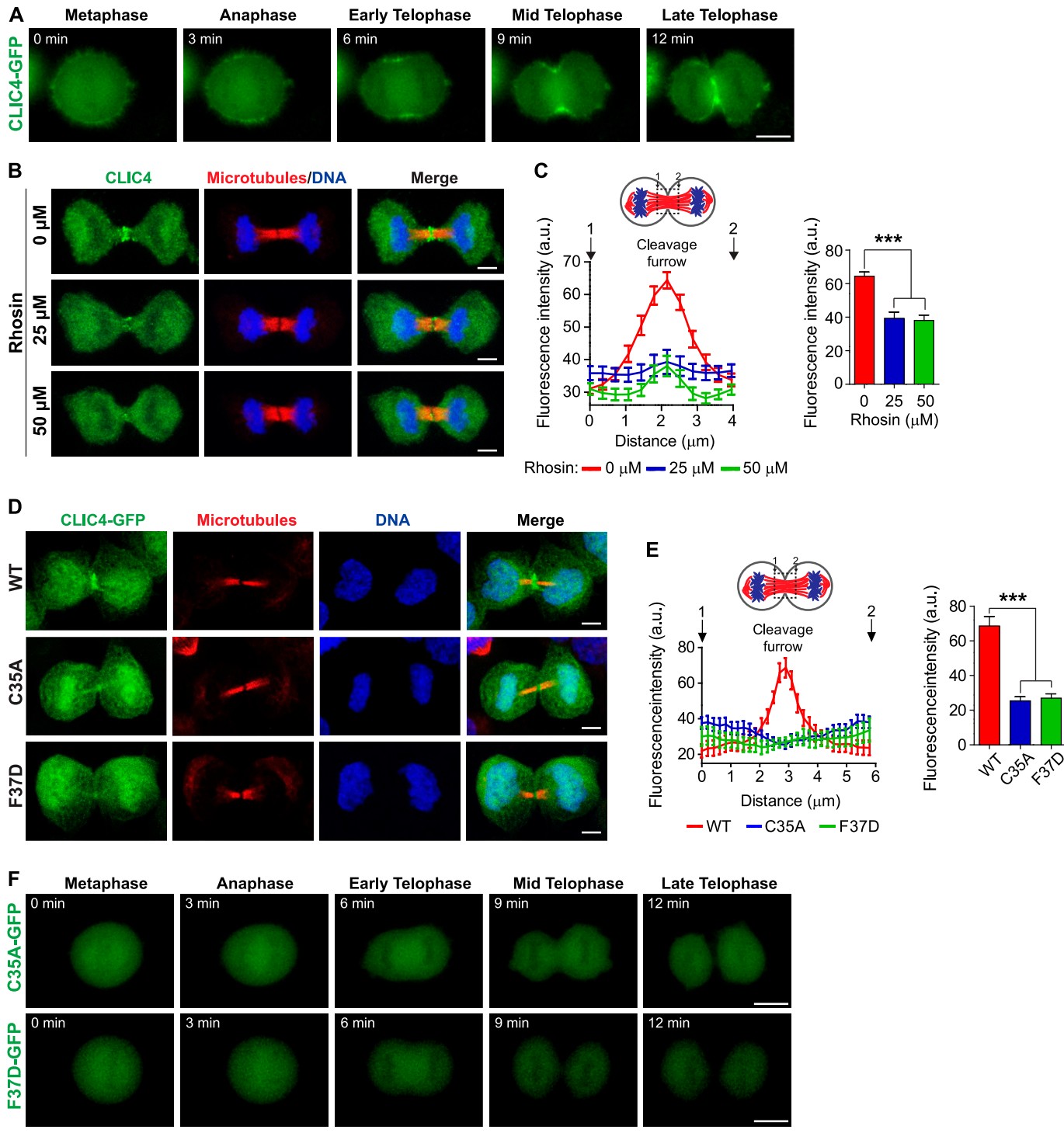

**Figure 1. The dynamic localization pattern of CLIC4 during cell division is dependent on RhoA activation and requires Cys35 and Phe37 residues.**
**(A)** Live-cell imaging of CLIC4 translocation to the cleavage furrow. Imaging of stable CLIC4-GFP expressing cells was performed. Frames from a time-lapse movie at the indicated stages are shown. Scale bar, 10 μm. **(B)** Representative images of CLIC4 localization at the cleavage furrow/midbody after RhoA inhibition. 0, 25, and 50 μM concentrations of Rhosin (RhoA inhibitor) were used to decrease the activity of intracellular RhoA during cytokinesis. Maximum intensity projections of Z-stacks show endogenous CLIC4 (*green*), microtubules (β-tubulin, *red*), and DNA staining (DAPI, *blue*). Scale bar, 5 μm. **(C)** Quantification of CLIC4 localization at the cleavage furrow after RhoA inhibition in 0 μM (*red trace*, *n* = 15 cells), 25 μM (*blue trace*, *n* = 15 cells), and 50 μM Rhosin-treated cells (*green trace*, *n* = 18 cells). One-way ANOVA with Dunnett's post hoc test was performed to compare endogenous CLIC4 intensities at the cleavage furrow (*right*) (***$P < 0.001$). Mean ± SEM are shown. **(D)** Representative images of CLIC4 localization at the cleavage furrow/midbody in stable wild-type (WT) and mutant (C35A and F37D) CLIC4-GFP expressing cells. Maximum intensity projections of Z-stacks show GFP-tagged WT and mutant CLIC4 proteins (*green*), microtubules (α-tubulin, *red*), and DNA staining (DAPI, *blue*). Scale bar, 5 μm. **(E)** Quantification of wild-type and mutant CLIC4-GFP localizations at the cleavage furrow. GFP intensities at the cleavage furrow were measured for CLIC4-GFP WT

proteins around 72–75 kD (Niers et al, 2011) and residual mycbirA*-CLIC4 proteins were eluted from streptavidin beads (Fig 2C, left panels, Elu (elution) fractions). With supplemental biotin, both WT and mutant mycBirA*-CLIC4 proteins were able to biotinylate multiple proteins, and these biotinylated proteins were successfully purified by streptavidin pull-down (Fig 2C, top right panel, whole cell lysate [WCL] and Elu fractions). MycBirA*–fused proteins were also monitored by anti-CLIC4 blotting to confirm their ectopic expression (Fig 2C, bottom panels).

To identify proximal interactors of CLIC4, the biotinylated proteins were isolated by streptavidin pull-down, then subjected to the on-bead tryptic digestion followed by liquid chromatography-tandem mass spectrometry (LC–MS/MS) analysis. Untransfected monopolar cytokinesis cells incubated with biotin were used as control. The experiment was conducted twice and only the proteins identified in both biological replicates, but not in the control, were listed (Table S1). CLIC4 was found as the top hit with 67% and 61% coverages for WT and C35A mutant, respectively. 85 proteins were identified in the wild-type CLIC4 interactome in total and 54 of them were specific to the wild type (Fig 2D and Table S1). The mutant CLIC4 interactome comprised less plasma membrane proteins compared with the wild type, most probably because of the abolished plasma membrane and cleavage furrow localization of CLIC4-C35A. Protocadherin alpha-9 (PCDHA9), SLC7A5, SLC13A3, SLC25A3, GP1BB, FLNB, ezrin (EZR), and ALIX were among the many plasma membrane proteins identified as wild-type CLIC4-specific proximity interactors in cytokinesis (Fig 2D).

## CLIC4 associates with ezrin, anillin, and ALIX during cell division

After the identification of CLIC4 proximity interactors in monopolar cytokinesis cells by mass spectrometry analysis, we focused on three proteins that were required for cytokinesis and appeared in the wild-type CLIC4-specific interactome: ezrin, anillin, and ALIX. Activated ezrin anchors the cortical actomyosin network to the plasma membrane as the cell progresses into mitosis and accumulates at the cleavage furrow upon chromosome segregation (Bretscher et al, 2002; Ramkumar & Baum, 2016). Anillin directly interacts with RhoA and shows a similar localization pattern as RhoA at the cleavage furrow. It is suggested to act as a scaffold protein connecting RhoA with the cortical ring components actin and myosin (Piekny & Glotzer, 2008). Anillin is also involved in the transition of the contractile ring into the midbody ring by both constituting the midbody structure and ensuring its plasma membrane linkage (Kechad et al, 2012). ALIX functions at the abscission step of cytokinesis, and its interactions with CEP55 and ESCRT-III are essential for a successful cell division (Morita et al, 2007). As all the mentioned proteins contribute to cytokinesis at different levels, we decided to dissect their interaction with CLIC4 to unravel its function in cell division.

To address whether CLIC4 cooperates with anillin, ezrin, and ALIX during cytokinesis, we first examined their localization in dividing cells. Both CLIC4 and anillin accumulated at the late furrow ingression region. Ezrin and CLIC4 showed a similar co-

localization pattern as the anillin–CLIC4 pair. ALIX concentrated at the midbody along with CLIC4 at the late telophase (Fig 3A). Given its co-localization with these proteins, we tested whether CLIC4 interacts physically with anillin, ezrin, and ALIX during cytokinesis. Co-immunoprecipitation using the sensitive GFP-trap approach indicates that CLIC4, ezrin, and ALIX are in the same complex interacting with each other either directly or indirectly, and the absence of anillin in this complex might imply its transient or weak interaction with CLIC4 (Fig 3B). To further verify their interactions in intact cells, we applied the proximity ligation assay (PLA) (Soderberg et al, 2006). For this, we used CLIC4-anillin, CLIC4-ALIX, and CLIC4-ezrin antibody pairs and analyzed the protein–protein interactions at the spatial resolution during cytokinesis. As expected, either none or a few PLA signals were detectable in control cells treated with only one primary antibody (Figs 3C and S2). The pairs of antibodies tested suggests the physical interactions of CLIC4 with anillin, ALIX, and ezrin in cytokinesis cells (Fig 3C). The number of PLA signals were highest for the interactions with ezrin and ALIX, which is in line with the result of the co-immunoprecipitation assay (Fig 3D). However, the detected interactions were not confined to the cleavage furrow. This may be due to other functions of CLIC4: CLIC4 localizes to early and recycling endosomes in epithelial cells where it inhibits the formation of branched actin on early endosomes (Argenzio et al, 2014; Chou et al, 2016). Ezrin activation is partly associated with recycling endosomes, and ALIX is also recruited to the endosomes as an ESCRT-associated protein (Matsuo et al, 2004; Dhekne et al, 2014). Therefore, the interactions of CLIC4 with ezrin and ALIX all over the cells may be due to the massive functioning of the endocytic machinery required for the remodeling of plasma membrane during cytokinesis (Horgan & McCaffrey, 2012).

## Knockouts of CLIC4 and CLIC1 lead to late cytokinesis defects and multinucleation

Because we identified that CLIC4 has a dynamic spatiotemporal localization during cell division and that it interacts with several proteins with critical functions in cytokinesis, we decided to more directly assess the role of CLIC4 in cytokinesis by using CRISPR/Cas9-mediated reverse genetics. To reduce off-target effects of editing, two different guide RNAs targeting the first and third exon of CLIC4 were generated, and control cells were treated with non-targeting guide RNA in parallel (Fig 4A). We examined the extent of cell division failures in these cells. The knockout of CLIC4 caused the cells to become multinucleated. Although 3.0% of the control cells had multinucleation, this percentage significantly increased to 10.23% and 7.43% in CLIC4 knockout cell lines (Fig 4C). CLIC4-GFP expression in CLIC4 knockout cells largely rescued the multinucleation phenotype, resulting in a significant decrease to 4.88% and 4.08% (Fig 4B and C).

As CLIC4 and CLIC1 share 67% sequence homology showing a high degree of structural similarity (Littler et al, 2005), and both were enriched at the mitotic cell surface in our proteomic analysis (Ozlu et al, 2015), CLIC1 caught our interest and we set out to determine if it cooperates with CLIC4 during cell division. Live-cell imaging of

(red trace, n = 7 cells), C35A mutant (blue trace, n = 7 cells), and F37D mutant (green trace, n = 7 cells) and compared by using one-way ANOVA with Dunnett's post hoc test (right) (***P < 0.001). Mean ± SEM are shown. **(F)** Live-cell imaging of stable mutant CLIC4-GFP–expressing cells during cell division, C35A mutant (top) and F37D mutant (bottom). Representative frames of time-lapse movies at the indicated stages are shown. Scale bar, 10 μm.

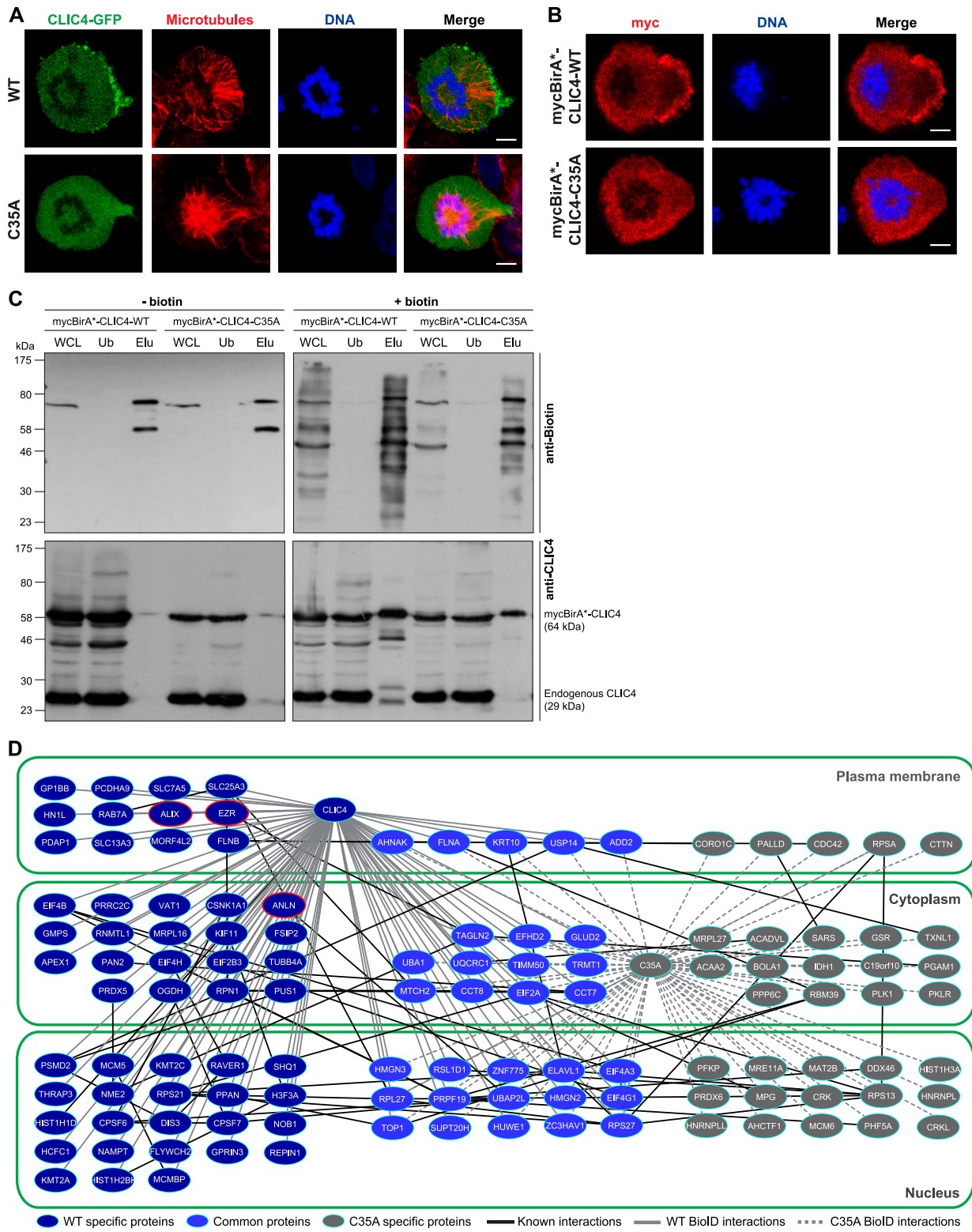

**Figure 2. Comparison of proximity interactomes reveals wild-type CLIC4–specific protein interactions in cytokinesis.**
**(A)** Subcellular localization of WT (*top*) and C35A-mutant (*bottom*) CLIC4-GFP proteins in cells arrested at monopolar cytokinesis. Images show GFP-tagged WT and C35-mutant CLIC4 proteins (*green*), microtubules (α-tubulin, *red*), and DNA staining (DAPI, *blue*). Scale bars, 5 µm. **(B)** Subcellular localization of mycBirA*-tagged WT and C35A-mutant CLIC4 proteins in monopolar cytokinesis cells. The cells were seeded on glass coverslips, transfected with mycBirA*-CLIC4-WT (*top*) and mycBirA*-CLIC4-C35A (*bottom*), and arrested at monopolar cytokinesis. Images show myc-tag (*red*) and DNA staining (DAPI, *blue*). Scale bars, 5 µm. **(C)** Affinity capture of biotinylated proteins using lysates of mycBirA*-CLIC4-WT and mycBirA*-CLIC4-C35A transfected monopolar cytokinesis cells either with (*right*) or without supplemental biotin (50 µM) (*left*).

CLIC1-GFP showed that it also localized to the cell surface in mitosis and accumulated at the cleavage furrow during cytokinesis (Fig S3A and Video 3). The immunostaining of both endogenous and GFP-tagged CLIC1 revealed a clear cleavage furrow localization in cytokinesis cells (Fig S3B and C). Our data indicate that CLIC1 exhibits a similar localization pattern as seen with CLIC4 during cell division (Fig S3D).

We generated CLIC1 knockout cell lines using the CRISPR/Cas9 system to probe its role in cell division. Two different sgRNAs targeting the first and second exon efficiently knocked out *CLIC1* and eliminated its expression (Fig 4D). We observed a significant increase in the multinucleation percentage of the CLIC1 knockout cells (11.48% and 6.66%) in comparison with the control cells (3.95%) (Fig 4F). The ectopic CLIC1-GFP expression in CLIC1 knockout cell lines caused the multinucleation percentages to drop to 4.02% and 3.97% by rescuing the phenotype (Fig 4E and F).

To investigate whether CLIC4 and CLIC1 act together during cytokinesis and have redundant roles in the progression of cytokinesis, we double knocked out CLIC4 and CLIC1 in HeLa S3 cells. We generated two different CLIC4 and CLIC1 double-knockout (hereafter, CLIC1/4 DKO) single cell colonies and confirmed the lack of CLIC4 and CLIC1 expression in these cell lines by Western blotting (Fig 4G). Double-knockout cells exhibited a markedly increased cell size with the development of polyploid cells having multiple nuclei and an increased multinucleation rate of 16%, which is significantly high compared with the control cells (5.07%) (Fig 4H). The fold change analysis of multinucleation percentages in single and double knockout cells revealed that multinucleation was higher but not significantly different in double-knockout cells (Fig S4A), indicating that there is no additive effect of double-knockout of CLIC1 and CLIC4 in the multinucleation condition of cells. This implies that CLIC1 and CLIC4 do not act independently from each other and are involved in the same pathway during cell division. To test whether CLIC4 and CLIC1 functionally complement each other, we overexpressed CLIC1-GFP in CLIC4 KO cells and vice versa by overexpressing CLIC4-GFP in CLIC1 KO cells. Although there was a slight change in the multinucleation rates in both cases, they were not statistically significant, suggesting that CLIC4 and CLIC1 cannot fully complement each other's function during cytokinesis (Fig S4B).

To characterize the defect leading to the multinucleated cell phenotype, we performed live-cell imaging of control and CLIC1/4 DKO cells. Although CLIC1/4 DKO cells were able to properly round up and accomplish cleavage furrow ingression and midbody formation as control cells, they were more prone to form multinucleated cells due to cleavage furrow regression in late cytokinesis (Fig 4I and J, Videos 4, and 5). The quantification of cell divisions revealed a significantly high percentage of cells with incomplete abscission in CLIC1/4 DKO cells (Fig 4I), which leads to the reunion of daughter cells and a significant increase in multinucleated cells. These results suggest that both CLIC4 and CLIC1 are required for the completion of cleavage furrow ingression and abscission during cytokinesis.

## CLIC4 and CLIC1 regulate membrane blebbing and accumulate at blebs during retraction

As CLIC proteins are suggested to be involved in actin-mediated processes, we attempted to explore the role of CLIC4 and CLIC1 in cortical actin dynamics during cell division. For this purpose, we applied time-lapse microscopy to image CLIC1/4 DKO cells expressing Lifeact-RFP, a marker to visualize F-actin in live cells (Riedl et al, 2008). Our data revealed abnormal blebbing at the poles in CLIC1/4 DKO cells compared with control cells (Fig 5A and Videos 6, and 7). Blebbing is a very fast protrusion of the plasma membrane that is frequently observed during cell migration, cytokinesis, and apoptosis. It is suggested that either the detachment of the plasma membrane from the cortical actin network or the rupture in the actin cortex results in bleb nucleation (Charras, 2008). The polar blebs were shown to be important for the release of intracellular pressure due to the constriction of the actomyosin ring at the equator during cytokinesis (Sedzinski et al, 2011). Although membrane blebbing is a physiological phenomenon, the extent of protrusions was significantly higher in CLIC1/4 DKO cells than in control cells (Fig 5B). This result implies the involvement of CLIC proteins in polar relaxation via blebbing upon constriction of the contractile ring. To further support this, we monitored dynamic localization patterns of GFP-tagged CLIC proteins at the polar blebs during cytokinesis. Our time-lapse data revealed rapid bleb expansions at anaphase that initially did not involve CLIC proteins. The gradual accumulation of CLIC proteins at the bleb cortex was then coupled with the bleb retraction. In the end, the strong cumulative signal of CLIC proteins dissolved with the disappearance of blebs (Fig 5C and Videos 8, and 9). Collectively, these data suggest that CLIC proteins are involved in the regulation of polar blebs during cytokinesis.

## A positive feedback loop regulates ezrin phosphorylation and cleavage furrow localization of CLIC4

Our data revealed the direct interaction between CLIC4 and ezrin during cytokinesis (Fig 3A–D). Ezrin is a member of the ERM family and acts as a linker between the plasma membrane and the underlying cortical actin cytoskeleton (Tsukita et al, 1997). To further assess whether CLIC4 and ezrin function together during cell division, we performed live-cell imaging of HeLa S3 cells stably expressing CLIC4-GFP and ezrin-RFP. They first co-localized at the

Whole cell lysate (WCL) and streptavidin pull-down samples (unbound [Ub] and elute [Elu] fractions) were immunoblotted with anti-biotin (*top*) and anti-CLIC4 antibodies (*bottom*). mycBirA*-CLIC4-WT and mycBirA*-CLIC4-C35A were able to biotinylate proteins and the biotinylated proteins were purified by streptavidin pull-down. **(D)** Comparison of WT and C35A-mutant CLIC4 proximity interactomes in monopolar cytokinesis. Nodes represent unique proteins identified and categorized according to their presence in only one network (WT-specific proteins, *dark blue nodes*; C35A specific proteins, *grey nodes*) or in both networks (common proteins, *light blue nodes*). Gene Ontology cellular component analysis was performed to group proteins based on their subcellular localization (*nucleus*, *cytoplasm*, and *plasma membrane*). Black edges represent already known protein–protein interactions within the interactomes determined by the STRING v10.5 database. Solid grey and dashed grey edges represent WT and C35A-mutant specific BioID interactions of CLIC4, respectively. Nodes highlighted with red lines (ezrin [EZR], anillin [ANLN], and ALIX) represent WT-specific interaction partners of CLIC4 selected for further analysis.

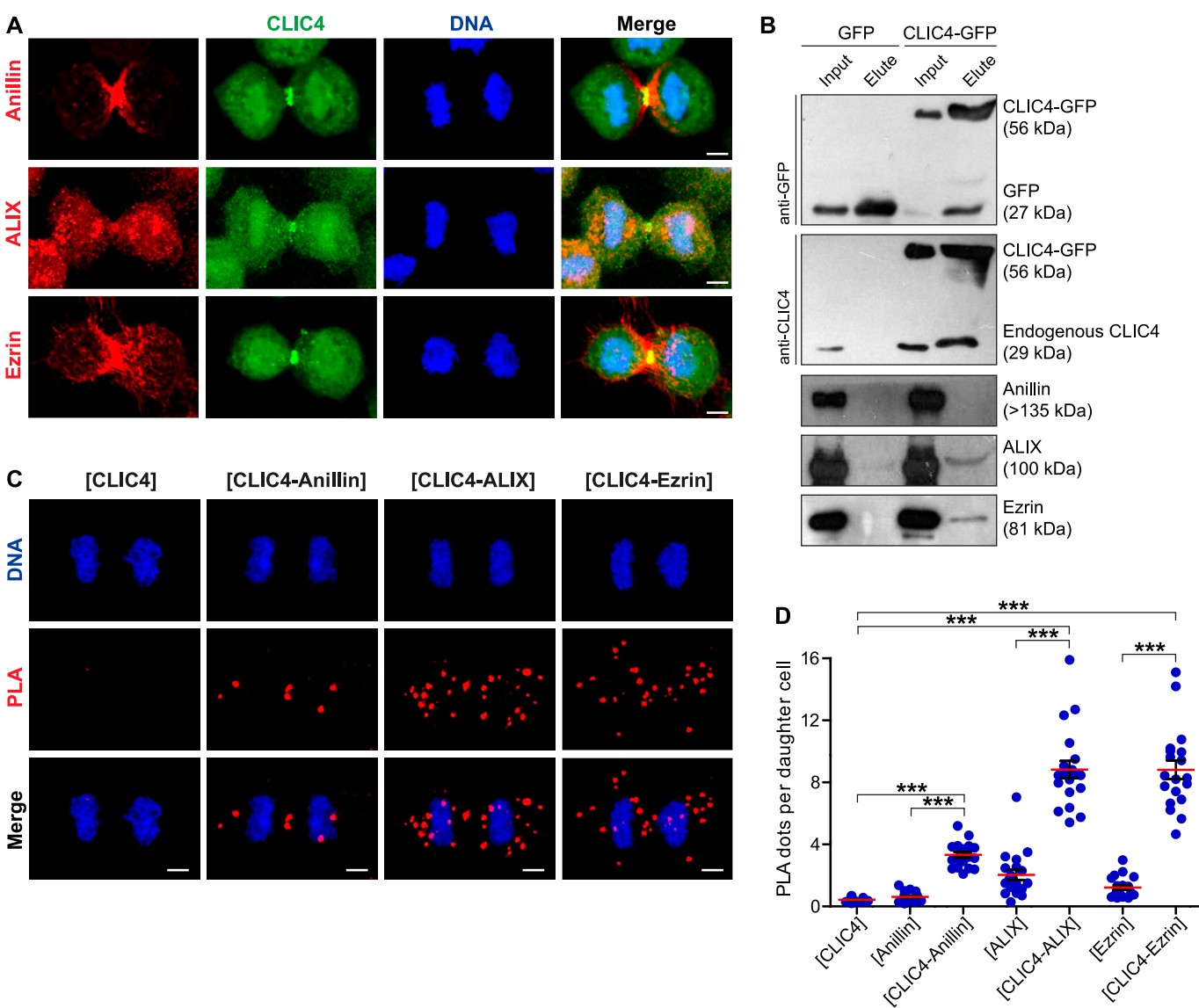

**Figure 3. CLIC4 interacts with anillin, ALIX, and ezrin in cytokinesis.**
**(A)** Co-localization of endogenous CLIC4 with anillin, ALIX, and ezrin in cytokinesis. Maximum intensity projections of Z-stacks show endogenous anillin, ALIX, and ezrin (*red*), endogenous CLIC4 (*green*), and DNA staining (DAPI, *blue*). Scale bars, 5 μm. **(B)** Co-immunoprecipitation analysis of interaction partners of CLIC4 in cytokinesis arrested cells using the sensitive GFP-trap approach. Western blot analyses of whole cell lysates (*input*) and elute fractions obtained from stable GFP (as control) and CLIC4-GFP expressing cells were performed against anti-GFP (*top*), anti-CLIC4 (*middle*), anti-anillin, anti-ALIX, and anti-ezrin antibodies (*bottom*). Co-immunoprecipitation of ALIX and ezrin with only CLIC4-GFP indicates that CLIC4 resides in the same complex with ezrin and ALIX during cytokinesis. **(C)** Spatial analysis of the interactions of CLIC4 with anillin, ALIX, and ezrin in intact cytokinesis cells by in situ proximity ligation assay (PLA). The representative images show the interactions between the examined antibody pairs as red fluorescent PLA puncta. Cells were also stained with DAPI for DNA (*blue*). Control was treated with only anti-CLIC4 antibody. Each image is the maximum intensity projection of a Z-stack and represent a typical cell staining observed in 20 fields chosen randomly. Scale bars, 5 μm.
**(D)** Quantification of the interactions of CLIC4 in cytokinesis by in situ PLA. In addition to only anti-CLIC4 antibody–treated cells (*n* = 533), cells subjected to only anti-anillin (*n* = 639), anti-ALIX (*n* = 538), and anti-ezrin (*n* = 499) antibodies (Fig S2) were also quantified as controls. The cells treated with antibody pairs of CLIC4-anillin (*n* = 757), CLIC4-ALIX (*n* = 456), and CLIC4-ezrin (*n* = 499) were used for the interaction analysis. Red lines indicate the mean number of PLA dots per daughter cell nucleus. Comparison of multiple groups was performed by one-way ANOVA with Bonferroni post hoc test (***$P < 0.001$). Mean ± SEM are shown.

mitotic cell surface and then were enriched at the equatorial cortex where the cleavage furrow started to form. In telophase, both CLIC4 and ezrin highly concentrated at the cleavage furrow (Fig 6A and Video 10).

All members of the ERM family have an F-actin–binding domain at their carboxyl termini and support actin cytoskeleton rearrangements in their active state (Turunen et al, 1994). The activation of ezrin requires a conformational switch regulated by PIP$_2$ interaction at the plasma membrane and phosphorylation of a conserved threonine (T567) located in the carboxyl terminus (Bosk et al, 2011). Previously, the interaction of the active form of ezrin with CLIC4 was shown (Viswanatha et al, 2013). To test if activation of ezrin is required for the cleavage furrow localization of CLIC4, we used an ezrin inhibitor, NSC305787. This drug inhibits the T567 phosphorylation of ezrin by

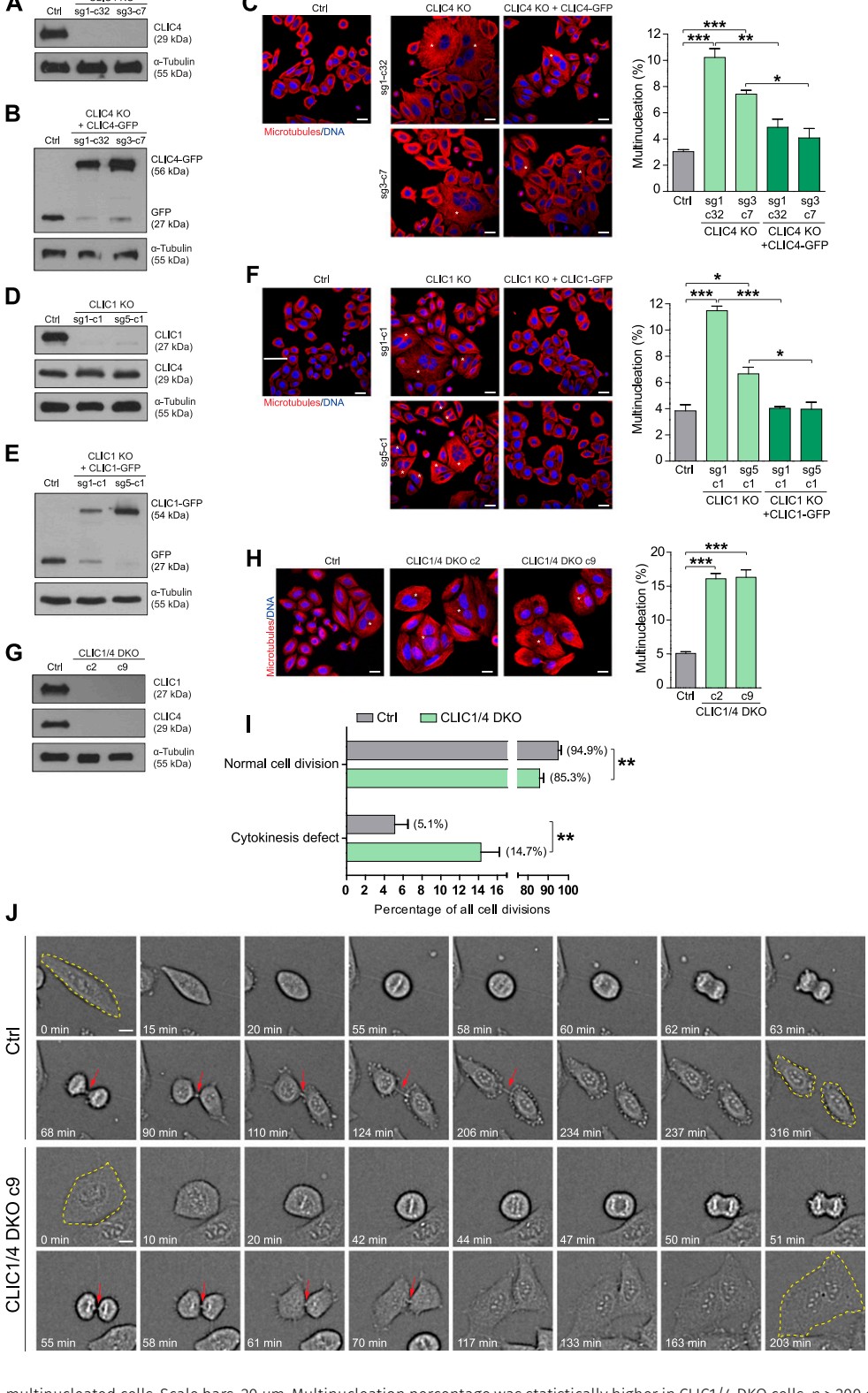

**Figure 4. Single and double knockouts of CLIC4 and CLIC1 lead to multinucleated cells.**

**(A)** CLIC4 knockout (KO) validation was achieved by Western blot analysis of the whole cell lysates using anti-CLIC4 antibody. α-Tubulin was used as a loading control. **(B)** Western blot analysis of CLIC4-GFP expression in the CLIC4 KO rescue cell lines. Whole cell lysates were immunoblotted with anti-GFP antibody to detect the expression of CLIC4-GFP. α-Tubulin was used as a loading control. **(C)** Quantification of multinucleation percentage of CLIC4 KO and CLIC4-GFP expressing CLIC4 KO rescue cell lines. Cells were stained with anti-α-tubulin antibody (*red*) and DAPI for DNA (*blue*). Asterisks in the representative images denote multinucleated cells. Scale bars, 20 μm. Multinucleation percentage was statistically higher in CLIC4 KO cells, and CLIC4-GFP expression in CLIC4 KO cells significantly decreased multinucleation percentage. $n ≥ 300$ cells per experiment were quantified and mean ± SEM of three independent experiments are shown. One-way ANOVA with Bonferroni post hoc test was used for statistical analysis (*$P <$ 0.05; **$P <$ 0.01; ***$P <$ 0.001). **(D)** CLIC1 KO validation was achieved by Western blot analysis of the whole cell lysates using anti-CLIC1 antibody. CLIC4 expression was not altered in CLIC1 KO cells as revealed by immunoblotting with anti-CLIC4 antibody. α-Tubulin was used as a loading control. **(E)** Western blot analysis of CLIC1-GFP expression in the CLIC1 KO cell lines. Whole cell lysates were immunoblotted with anti-GFP antibody to detect expression of CLIC1-GFP. α-Tubulin was used as a loading control. **(F)** Quantification of multinucleation percentage of CLIC1 KO and CLIC1-GFP expressing CLIC1 KO rescue cell lines. Cells were stained with anti-α-tubulin antibody (*red*) and DAPI for DNA (*blue*). Asterisks in the representative images denote multinucleated cells. Scale bars, 20 μm. Multinucleation percentage was statistically higher in CLIC1 KO cells and CLIC1-GFP re-expression in CLIC1 KO cells significantly decreased multinucleation percentage. $n ≥ 300$ cells per experiment were quantified and mean ± SEM of three independent experiments are shown. One-way ANOVA with Bonferroni post hoc test was used for statistical analysis (*$P <$ 0.05; **$P <$ 0.01; ***$P <$ 0.001). **(G)** CLIC1/4 double KO (DKO) validation was achieved by Western blot analysis of the whole cell lysates using anti-CLIC4 and anti-CLIC1 antibodies. α-Tubulin was used as a loading control. **(H)** Quantification of multinucleation percentage of CLIC1/4 DKO cell lines. Cells were stained with anti-α-tubulin antibody (*red*) and DAPI for DNA (*blue*). Asterisks in the representative images denote

multinucleated cells. Scale bars, 20 μm. Multinucleation percentage was statistically higher in CLIC1/4 DKO cells. $n ≥ 200$ cells per experiment were quantified and mean ± SEM of three independent experiments are shown. One-way ANOVA with Dunnett's post hoc test was used for statistical analysis (***$P <$ 0.001). **(I)** Quantification of successful and defective cell divisions of control and CLIC1/4 DKO cells by live-cell imaging. Cells were imaged for 24 h. $n =$ 371 cell divisions from six experiments for control cells and $n =$ 526 cell divisions from eight experiments for CLIC1/4 DKO cells were quantified and mean ± SEM are shown. Unpaired two-tailed *t* test was used for significance analysis (**$P <$ 0.01). **(J)** Representative still images of time-lapse movies showing a successful cell division in control cell line and cytokinesis defect in CLIC1/4 DKO c9 cell line. Dividing cells were marked with yellow dashed lines before and after cell division. Red arrows indicate the presence of midbody. Scale bars, 10 μm.

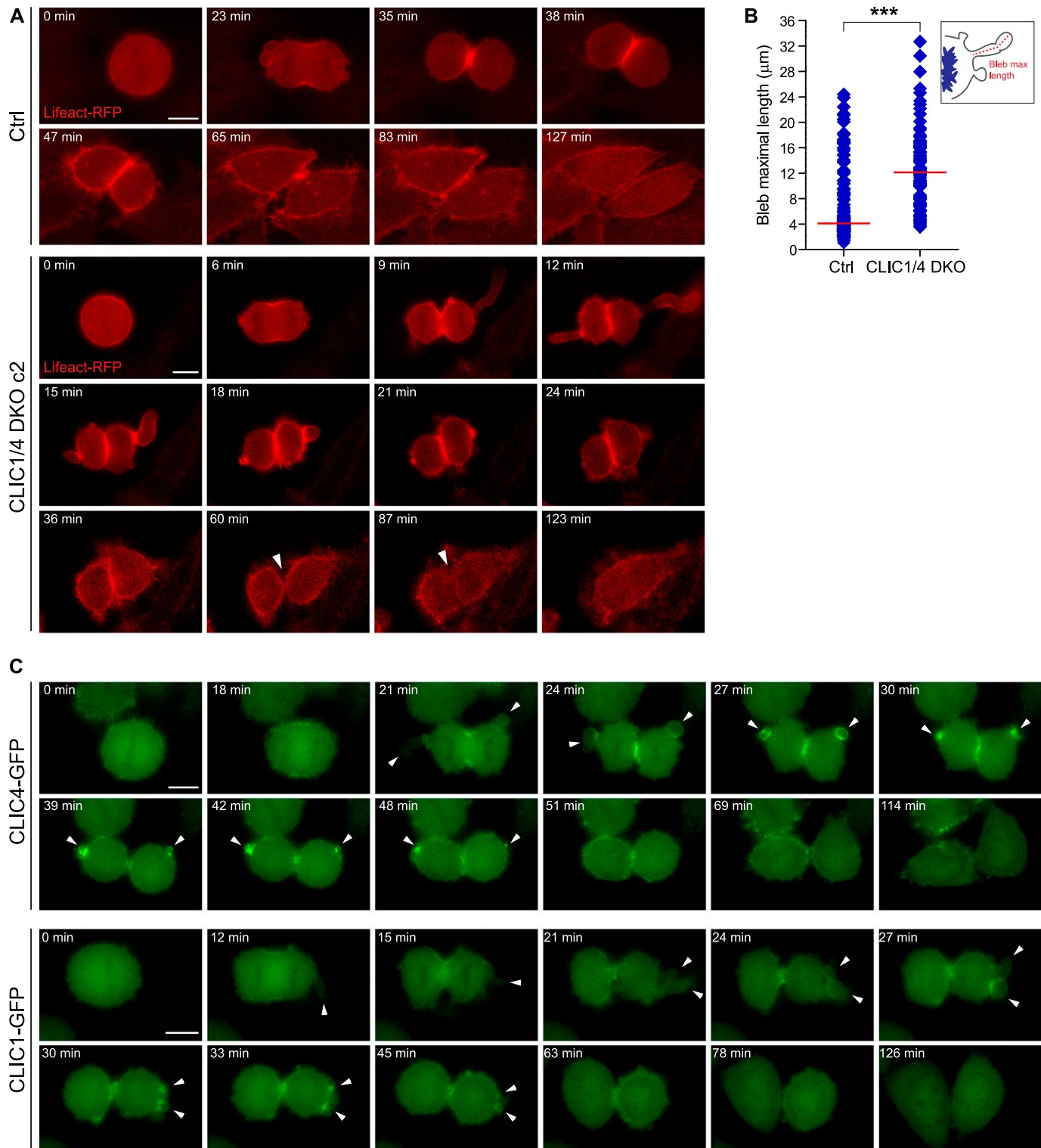

**Figure 5. CLIC4 and CLIC1 regulate membrane blebbing during cytokinesis.**
**(A)** Live-cell imaging of stable Lifeact-RFP expressing control (*top*) and CLIC1/4 DKO c2 (*bottom*) cells. Representative frames of time-lapse movies at the indicated time points are presented. CLIC1/4 DKO cells show abnormal bleb formation at the polar cortex during cell division. Arrowheads denote the furrow regression in CLIC1/4 DKO cells and so the multinucleated cell formation. Scale bar, 10 μm. **(B)** Quantification of maximal bleb extension during cytokinesis in control and CLIC1/4 DKO cells. *n* = 129 cells for control and *n* = 95 cells for CLIC1/4 DKO c2 were quantified and individual blebs were plotted based on their maximal length. Red lines indicate the median of bleb maximal length for each condition. Unpaired two-tailed *t* test was used for significance analysis (\*\*\**P* < 0.001). **(C)** Spatiotemporal regulation of CLIC4-GFP (*top*) and CLIC1-GFP (*bottom*) during bleb retraction. Representative frames of time-lapse movies at the indicated time points are presented. Arrowheads denote membrane blebs at different phases, from expansion to retraction during cytokinesis. Both CLIC4-GFP and CLIC1-GFP localize to the bleb rim during retraction phase of the membrane blebs. Scale bar, 10 μm.

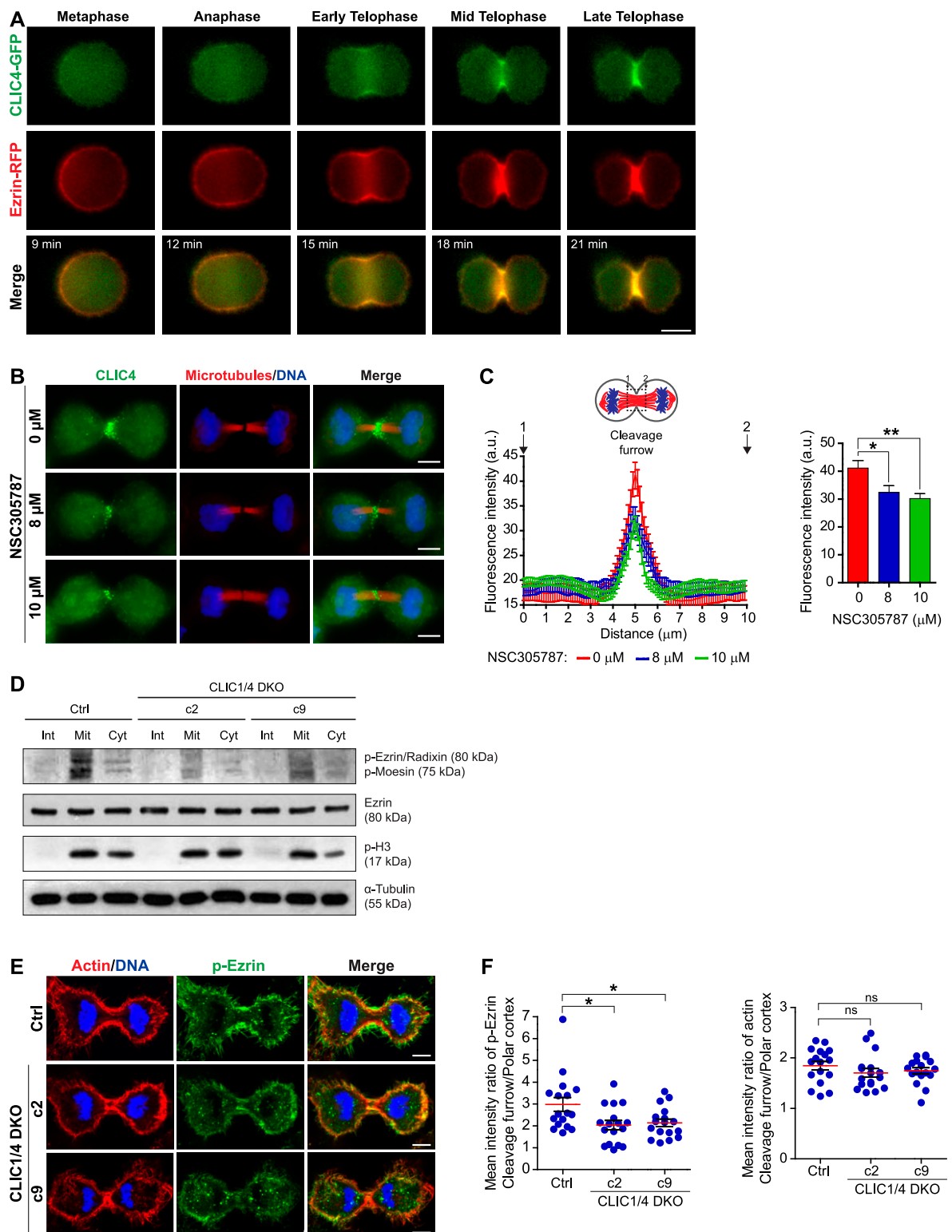

**Figure 6.  CLIC4 enhances ezrin phosphorylation to be translocated to the cleavage furrow.**
**(A)** Live-cell imaging of CLIC4-GFP and ezrin-RFP translocation to the cleavage furrow. Imaging of stable CLIC4-GFP and ezrin-RFP expressing cells were performed and representative frames of a time-lapse movie at the indicated stages are shown. Scale bar, 10 μm. **(B)** Representative images displaying the effect of ezrin inhibition on CLIC4 localization at the cleavage furrow. 0, 8, and 10 μM concentrations of NSC305787 (ezrin inhibitor) were used to decrease the activity of intracellular ezrin during cytokinesis. Maximum intensity projections of Z-stacks show endogenous CLIC4 (*green*), microtubules (β-tubulin, *red*), and DNA staining (DAPI, *blue*). Scale bar, 10 μm. **(C)** Quantification of CLIC4 localization at the cleavage furrow in 0 μM (*red trace, n = 26 cells*), 8 μM (*blue trace, n = 21 cells*), and 10 μM NSC305787-treated cells (*green trace,*

protein kinase C iota type, thus actin binding of endogenous ezrin is abolished without altering cellular ezrin levels (Bulut et al, 2012). Strikingly, the inhibition of ezrin activation significantly decreased the CLIC4 signal at the cleavage furrow in a dose-dependent manner (Fig 6B and C). This implies that active ezrin is involved in the recruitment of CLIC4 to the cleavage furrow.

A previous study suggested that CLIC4 stimulates ERM activation in glomerular capillary endothelial cells (Tavasoli et al, 2016). To test whether CLIC4 plays a role in the activation of ERM during cytokinesis, we biochemically monitored phospho-ERM levels in control and CLIC1/4 DKO cells in a cell cycle–dependent manner by Western blotting. A burst of ERM activation observed in mitotic control cells was drastically reduced in knockout cells. Similarly, there is a prominent decrease in the phospho-ERM levels in knockout cells during cytokinesis (Fig 6D). To examine the ezrin activation spatially at the cleavage furrow, we imaged ezrin phosphorylation in CLIC1/4 knockout cells. Although phosphorylated ezrin significantly accumulated at the cleavage furrow in control cells, this was significantly reduced in knockout cells (Fig 6E and F, left panel). This reduction is not due to impaired actin dynamics at the cleavage furrow. The ratio of actin levels at the cleavage furrow and polar cortex did not show any difference between knockout and control cells (Fig 6F, right panel). Altogether, these results suggest that the cooperation of CLIC4 and ezrin during cytokinesis have mutual effects. The interaction of CLIC4 and ezrin stimulates ezrin activation, which facilitates the translocation of CLIC4 to the cleavage furrow.

## Discussion

CLIC proteins do not act as traditional chloride channels in cells in contrast to their nomenclature. They are metamorphic proteins and can adopt multiple reversible structures, which may be linked to their remarkable range of functions in cells from cellular differentiation to membrane trafficking (Jiang et al, 2014; Argenzio & Moolenaar, 2016). In this study, we examined CLIC proteins in the context of cell division and concluded that both CLIC4 and CLIC1 function in mammalian cytokinesis: In their absence, daughter cells fail to undergo the abscission stage of cytokinesis, which leads to a significant enrichment of multinucleated cells.

The extent of the cytokinesis failure in CLIC4 and CLIC1 knockout cells is modest, but it is likely that this is due to the two layers of redundancy in this process: the redundant pathways of cytokinesis (Eggert et al, 2006) and the functional redundancy of other CLIC family members (Littler et al, 2010). In line with our results, CLIC4 knockout mice were viable, but their phenotypes revealed modest abnormalities in cell division (Ulmasov et al, 2009). Particularly, the homozygote CLIC4 knockout embryos exhibited a decreased angiogenesis activity and a tendency to prenatal mortality. Their body weight was usually less than the wild-type or heterozygous ones. The reason for the mildness of the CLIC4 knockout mice phenotype has been proposed to be due to functional redundancy between CLIC4 and other CLICs (Ulmasov et al, 2009). Indeed in *C. elegans*, two members of CLICs (EXC4 and EXL1) can functionally complement each other (Berry & Hobert, 2006). To address the functional redundancy of CLICs during cell division in mammalian cells, we created CLIC4 and CLIC1 double knockout cells. However, the multinucleation rate did not drastically increase when compared with single knockouts (Fig S4A). In addition, overexpression of CLIC1 did not rescue the multinucleation phenotype of CLIC4 knockout cells and vice versa (Fig S4B). The differential expression pattern of CLIC4 and CLIC1 in distinct tissues and thus in cancer types suggests a variation in their functional roles (Peretti et al, 2015). Therefore, CLIC4 and CLIC1 may act during cytokinesis without complete functional complementation, and the redundant pathways of cell division may prevent complete failure of cytokinesis when both CLIC4 and CLIC1 are depleted.

In line with our previous quantitative proteomics study identifying CLIC proteins among the cell surface proteins enriched in mitosis (Ozlu et al, 2015), our microscopy-based data revealed that both CLIC4 and CLIC1 decorated mitotic cell surface and as cells progress into cytokinesis, they accumulated at the cleavage furrow and midbody arms. We showed that the cleavage furrow recruitment of CLIC4 depends on RhoA activation and requires the Cys35 and Phe37 residues. Based on the homology between CLIC4 and GST-omega 1, Cys35 is proposed as the critical active site for the putative enzymatic activity of CLIC4 and is found to be essential for its translocation to the plasma membrane (Ashley, 2003; Ponsioen et al, 2009). How does a single residue change the overall localization of the protein? One possible explanation for the cell cycle–dependent translocation of CLIC4 to the plasma membrane is its enzymatic activity. In this respect, the recent finding on the contribution of glutathione-dependent oxidoreductase activity of CLIC3 in cancer progression (Hernandez-Fernaud et al, 2017) provides physiological evidence for the enzymatic abilities of CLIC family. Another possibility for the regulation of CLIC4 translocation is through a posttranslational modification, namely, S-nitrosylation. Hypernitrosylation of CLIC4 in the Cys35 mutant, which affects its stability and nuclear residence (Malik et al, 2010, 2012), supports the importance of this Cys residue in its translocation. However, given the fact that the cell surface and cleavage furrow localizations of CLIC4 are also abolished in Phe37 mutants, another residue at the enzymatically active cleft (Ashley, 2003; Ponsioen et al, 2009), it is

*n* = 26 cells). One-way ANOVA with Dunnett's post hoc test was performed to compare endogenous CLIC4 intensities at the cleavage furrow (*right*). (*$P < 0.05$; **$P < 0.01$). Mean ± SEM are shown. **(D)** Western blot analysis of ezrin and phospho–ezrin–radixin–moesin (pERM) in CLIC1/4 DKO cells at different phases of the cell cycle. Whole cell lysates were immunoblotted with anti-phospho-ERM antibody to detect phosphorylated ERM and with anti-ezrin antibody to observe endogenous ezrin expression levels. Although expression of endogenous ezrin was not altered in CLIC1/4 DKO cells, its phosphorylation was significantly decreased because of the absence of CLIC4 and CLIC1. Anti-phospho-H3 (p-H3) and anti-α-tubulin antibodies were used as mitosis marker and loading control, respectively. **(E)** Representative images of phospho-ezrin (p-ezrin) at the cleavage furrow in control and CLIC1/4 DKO cells. Maximum intensity projections of Z-stacks show phospho-ezrin (*green*), actin filaments (Phalloidin, *red*), and DNA staining (DAPI, *blue*). Scale bars, 5 μm. **(F)** Quantification of the mean intensity ratio of cleavage furrow to polar cortex for p-ezrin (*left*) and actin filaments (*right*) levels in control (*n* = 17 cells) and CLIC1/4 DKO cell lines (*n* = 17 cells for both *c2* and *c9*). One-way ANOVA with Dunnett's post hoc test was performed (*$P < 0.05$; ns, nonsignificant). Mean ± SEM are shown.

unlikely that translocation is solely S-nitrosylation-dependent. Yet, it is clear that these critical residues at the enzymatically active cleft govern subcellular localization, binding partners, and more importantly functions of CLIC4. For this, we systematically compared Cys35 mutant (i.e., C35A) and wild-type CLIC4 interactomes to identify the binding partners of CLIC4 required for its proper localization during cytokinesis. Our proteomics study identified ezrin, anillin, and ALIX as cytokinesis-specific interaction partners of CLIC4.

How do CLIC proteins contribute to cytokinesis? Our model suggests that CLIC4 may function as a docking protein in a plasma membrane–actin cytoskeleton anchoring complex via its association with scaffold and linker proteins, namely, anillin and ezrin in a RhoA activation–dependent manner during cytokinesis (Fig 7A). Although RhoA plays a central role in the translocation of CLIC4 to the cleavage furrow, we could not detect an interaction between them as a result of our proximity-dependent proteomics analysis. It is possible that the RhoA and CLIC4 interaction is transient, so it could not be captured in our assay. Another explanation could be their distal association in a complex, where anillin mediates the interaction between RhoA and CLIC4.

CLIC4 may not only be involved in the tethering of the cortical actin network with the plasma membrane but also with the spindle microtubules as anillin. Anillin was shown to link the spindle microtubules with the actomyosin contractile ring at the cleavage furrow via its interaction with a component of the centralspindlin complex, RacGAP50C (D'Avino et al, 2008). From this perspective, the identification of tubulin and KIF11, a plus end–directed kinesin that is essential for the spindle formation (Waitzman & Rice, 2014), as CLIC4 wild type–specific binding partners support this possibility. CLIC4 and CLIC1 do not only localize to the cleavage furrow but also to the midbody. The constriction of the contractile ring, which drives the formation of the midbody by compacting the central spindle (D'Avino & Capalbo, 2016), may facilitate the interaction of CLIC4 with the midbody component ALIX (Morita et al, 2007) and the spindle. RhoA-dependent interaction of CLIC4 with all these proteins may provide a bridge between actin filaments and the plasma membrane as well as with the microtubules of the midbody during the abscission stage (Fig 7A). The molecular details of this model remain to be solved.

CLIC4 is involved in the RhoA-mDia2 signaling pathway that regulates the cortical actin network via its direct binding partner profilin-1 (Argenzio et al, 2018). As a G-actin–binding protein, profilin-1 is known to promote the elongation of formin-nucleated linear actin filaments at the cortex and to suppress Arp2/3-nucleated branched actin assembly during cell division in fission yeast (Suarez et al, 2015). CLIC4 was also found to inhibit branched actin formation on early endosomes via its interaction with cortactin (Chou et al, 2016). These data suggest that CLIC4 might translocate to the cell surface upon RhoA activation in favor of filamentous actin generation over branched actin formation to increase cortical rigidity. Although we could not detect any pronounced difference in actin levels at the cortex, we observed massive blebbing at the poles in CLIC1/4 double-knockout cells that supports the role of CLIC proteins in the attachment of cortical actin filaments to the plasma membrane (Fig 7B). Similar abnormal cortical blebbing has been reported for the depletion of moesin,

the only member of the ERM family in *Drosophila* (Roubinet et al, 2011). Strikingly, our data revealed that both CLIC4 and CLIC1 localize to the cortical blebs as they retract, which is reminiscent of the recruitment of ezrin to the blebs (Charras et al, 2006). Collectively, these data strongly support that CLICs and ERMs function as a complex in regulating cortical stability. During cell elongation, the cortical rigidity is differentially regulated, which couples equatorial constriction and polar relaxation to execute cytokinesis (Ramkumar & Baum, 2016). The depletion of CLICs impairs the cortical dynamics, thus causing abnormal blebbing and cleavage furrow regression during cytokinesis.

How is the interaction of ezrin and CLIC4 regulated during cytokinesis? The phosphorylation of ezrin is known to stimulate its F-actin–binding (Bosk et al, 2011) and CLIC proteins have been reported to phosphorylate ERM proteins in glomerular endothelial cells, which is important for the cytoskeletal binding of ERMs and thereby the maintenance of normal glomerular capillary loop structure (Tavasoli et al, 2016). In good agreement with previous findings, we showed that the knockout of CLICs reduces the levels of ezrin phosphorylation at the cleavage furrow. Moreover, our data demonstrated that the inhibition of ezrin phosphorylation compromised the recruitment of CLIC4 to the cleavage furrow, suggesting a reciprocal regulation between ezrin and CLIC4 during cytokinesis (Fig 7C). RhoA activation is shown to stimulate the phosphorylation and translocation of ERM proteins to the membrane (Shaw et al, 1998). Similarly, our data revealed the re-localization of CLIC4 to the cleavage furrow upon RhoA activation. Collectively, these findings lead us to conclude that RhoA activation promotes ezrin activation in a CLIC4-dependent manner and stimulates co-translocation of CLIC4 and ezrin to the cortical actin network and plasma membrane interface. Further work is required to determine the exact molecular action of CLIC4 on ERM phosphorylation. In addition, the necessity of ezrin phosphorylation for the cleavage furrow/membrane localization of CLIC4 suggests their functional dependency in coupling the actomyosin cytoskeleton to the plasma membrane not only at the cleavage furrow but also at polar blebs for a successful cytokinesis.

Multiple lines of evidence from previous studies in various cancer cells proposed that CLIC4 functions as a tumor suppressor (Suh et al, 2012; Peretti et al, 2015). In addition, both CLIC4 and CLIC1 are overexpressed in cancer stem cells and inhibition of CLIC4 expression inhibits tumor growth (Suh et al, 2005; Wang et al, 2012; Deng et al, 2014; Peretti et al, 2015). Our study demonstrates the proliferative roles of CLIC4 and CLIC1 and how they improve the fidelity of cell division. The high conservation of CLICs across metazoans could be the consequence of their essential role in cell survival and maintenance of genomic integrity during cell proliferation. More work is required to decipher the relationship between the role of CLICs in cell cycle and cancer pathogenesis. Yet, our study supports the emerging potential of CLIC4 and CLIC1 as a therapeutic target in cancer progression.

In summary, our study attributes new functional aspects to CLIC proteins in cell division. The cleavage furrow/midbody-specific proximity interactors during cytokinesis may be potential "substrates" of the yet undefined enzymatic side of CLIC4. Further investigation of the molecular mechanisms behind the interaction of CLIC4 with ezrin, anillin, and ALIX during cytokinesis will provide

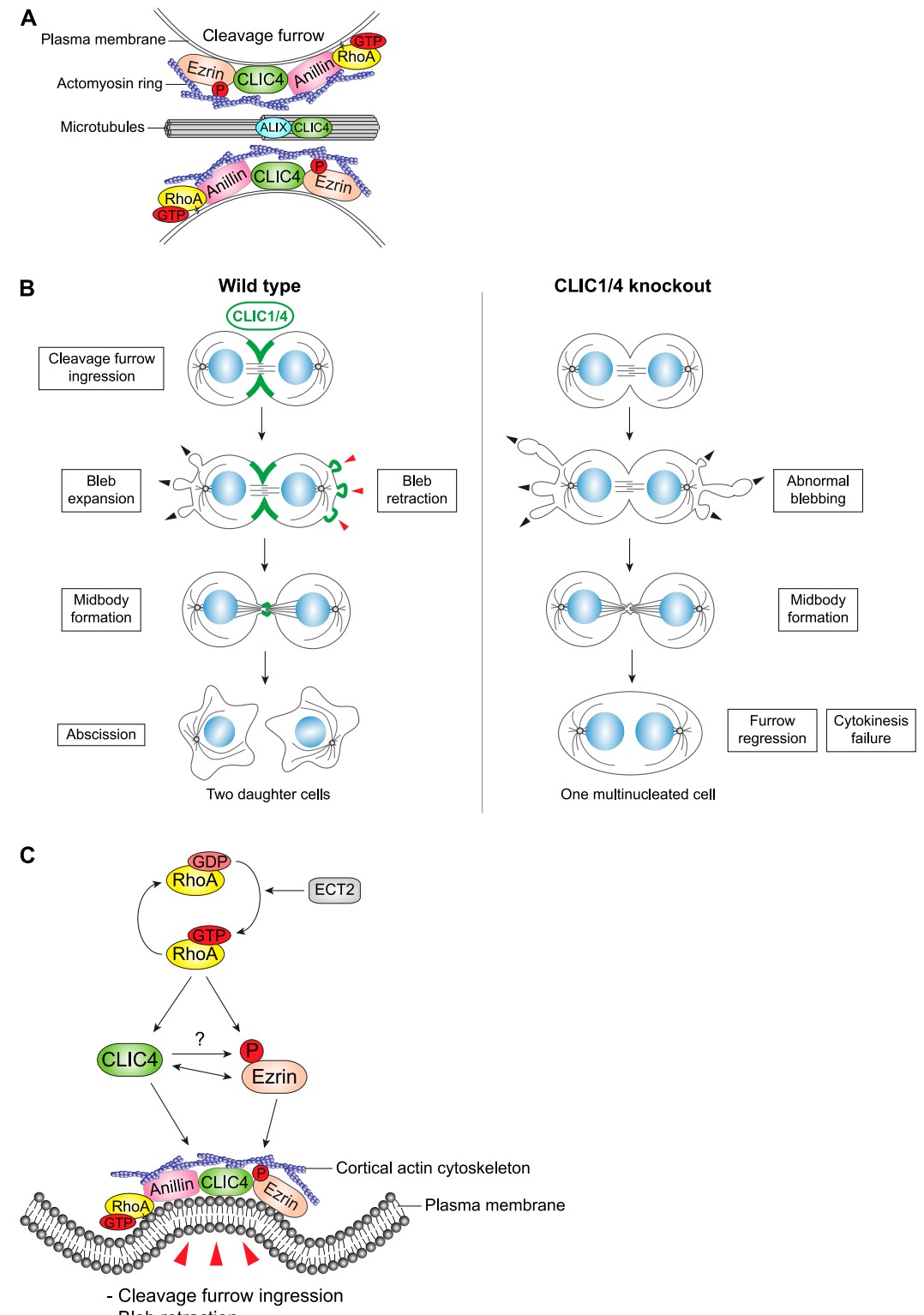

**Figure 7.  A model describing new roles of CLIC4 and CLIC1 during cytokinesis.**
**(A)** CLIC4 interacts with ezrin, anillin, and ALIX during cytokinesis and provides a link between RhoA signaling and actin cytoskeleton–plasma membrane anchorage at the cleavage furrow and midbody. **(B)** Both CLIC4 and CLIC1 localize to the cleavage furrow, polar blebs at the retraction phase, and midbody during cell division. The absence of CLIC4 and CLIC1 causes abnormal blebbing at the polar cortex and furrow regression at late cytokinesis which results in multinucleated cells. **(C)** The proposed signaling scheme of CLIC4 function in cleavage furrow ingression and bleb retraction. Activation of RhoA by its RhoGEF ECT2 leads to translocation of CLIC4 from cytosol to plasma membrane during cytokinesis, which promotes ezrin activation at the cleavage furrow through its phosphorylation. Reciprocally, ezrin activation facilitates CLIC4 accumulation at the cleavage furrow. This interaction regulates cortical stability by bridging actin cytoskeleton and plasma membrane during cleavage furrow ingression and bleb retraction.

important clues about its putative enzymatic activities and shed new light on animal cell cytokinesis.

## Materials and Methods

### Cell culture and synchronizations

HeLa S3 cells (ATCC CCL-2.2) were maintained in DMEM high glucose supplemented with 10% fetal bovine serum (Gibco) and 1% penicillin–streptomycin (Lonza) at 37°C in a humidified atmosphere containing 5% $CO_2$. The arrest of cell cycle progression during interphase was achieved by double thymidine block. Briefly, 2 mM thymidine (296542; Santa Cruz Biotechnology) was added to the cell culture medium, and the cells were incubated for 16 h at 37°C. Then, the cells were incubated in complete medium without thymidine for 8 h. After the second thymidine block of the cells for 16 h, most of the cells were arrested in the $G_1$ phase of interphase. For mitotic arrest, the cells were incubated in complete medium without thymidine for 8 h after the second thymidine block. Then, 10 ng/ml of nocodazole (487928; Calbiochem) was added to the cell culture medium and the cells were incubated for 5 h. At the end of the incubation with nocodazole, the cells were arrested in prometaphase. For cytokinesis arrest, the cells were incubated in complete medium for 1 h after nocodazole treatment. To induce monopolar cytokinesis, after the second thymidine block, the cells were incubated in complete medium containing 10 $\mu$M S-trityl-L-cysteine (STC) (164739; Sigma-Aldrich) for 16 h. Then, 100 $\mu$m purvalanol-A (1580; Tocris Bioscience) was added to the cell culture medium and the cells were incubated for 15 min.

### Oligonucleotide sequences and plasmids

The human CLIC4 and CLIC1 cDNAs were cloned into pEGFP-N1 plasmid (#6085-1; Clontech) to express GFP-tagged fusion proteins of CLIC4 and CLIC1. Site-directed mutagenesis was used to generate C35A and F37D mutations of CLIC4. CLIC4-WT, CLIC4-C35A, and CLIC4-F37D inserts amplified from pEGFP-N1 plasmids were cloned into pcDNA3.1 mycBioID (#35700; Addgene) to express mycBirA*-tagged BioID proteins (Roux et al, 2012). For the generation of stable cell lines, CLIC4-WT-GFP, CLIC4-C35A-GFP, CLIC4-F37D-GFP, and CLIC1-GFP inserts were cloned into pLenti CMV Hygro DEST (w117-1) plasmid (#17454; Addgene) by using Gateway cloning (Campeau et al, 2009). Lifeact-RFP and ezrin-RFP subcloned pLVX-Puro (Takara Bio Inc.) plasmids (Barry et al, 2015) were a kind gift from Dr Michael Way (The Francis Crick Institute, Lincoln's Inn Fields Laboratories, London WC2A 3LY, England, UK).

CRISPR/Cas9–based knockouts of CLIC4 and CLIC1 were performed using the following oligonucleotide sequences: human CLIC4 sgRNAs as top/bottom pairs, sg1 5'-CACCGGATGCCGCTGAATGGGCTGA-3'/5'-AAACTCAGCCCATTCAGCGGCATCC-3', sg3 5'-CACCGAACGGATGTAAATAA-GATTG-3'/5'-AAACCAATCTTATTTACATCCGTTC-3'; human CLIC1 sgRNAs as top/bottom pairs, sg1 5'-CACCGGCAGGTCGAATTGTTCGTGA-3'/5'-AAACTCACGAACAATTCGACCTGCC-3', sg5 5'-CACCGGTTCATGG-TACTGTGGCTCA-3'/5'-AAACTGAGCCACAGTACCATGAACC-3'; non-targeting sgRNA as top/bottom pair, 5'-CACCGACGGAGGCTAAGCGTCGCAA-3'/

5'-AAACTTGCGACGCTTAGCCTCCGTC-3'. All sgRNAs were cloned into lentiCRISPR v1 (#49535; Addgene) as described previously (Shalem et al, 2014).

### Transfection and lentiviral transduction

Cells were transfected using Lipofectamine 2000 transfection reagent (11668027; Invitrogen) according to the manufacturer's instructions. To produce lentiviruses, HEK293T cells were transfected with psPAX2 (#12260; Addgene) and pCMV-VSV-G (#8454; Addgene) (Stewart et al, 2003). The viral supernatants were collected at 48- and 72-h after transfection. HeLa S3 cells were infected with the pooled viral supernatants in medium supplemented with 2 $\mu$g/$\mu$l protamine sulfate (P4505; Sigma-Aldrich) as coadjutant. The virus-infected cells were selected by 2 $\mu$g/ml puromycin (P8833; Sigma-Aldrich) treatment for 2 d or 400 $\mu$g/ml hygromycin (H3254; Sigma-Aldrich) treatment for 1 wk. Single cell clones of CRISPR cell lines were obtained by serial dilution and then expanded in culture to obtain cell lines.

### Inhibitor treatments

Cells were treated with RhoA inhibitor, Rhosin (555460; Calbiochem), for 1 h after nocodazole treatment. 25 and 50 $\mu$M concentrations of Rhosin were used to decrease the activity of intracellular RhoA during cytokinesis. Ezrin inhibitor, compound NSC305787, was a kind gift from Dr Aykut Üren (Department of Oncology, Georgetown University Medical Center, Washington, DC, USA) (Bulut et al, 2012; Celik et al, 2015). During nocodazole treatment and nocodazole release, the cells were treated with 8 and 10 $\mu$M concentrations of NSC305787 for 6 h in total to inhibit ezrin activation. Control cells were treated with DMSO.

### Affinity capture of biotinylated proteins

HeLa S3 cells transfected with mycBirA*-CLIC4 plasmids were incubated in medium supplemented with 50 $\mu$M D-biotin (B-1595; Life Technologies) and 10 $\mu$M STC (164739; Sigma-Aldrich) for 12 h after second thymidine block. The cells were then arrested at monopolar cytokinesis by purvalanol-A treatment and collected by scraping. Then, the cells were lysed in the lysis buffer (50 mM Tris, pH 7.4; 500 mM NaCl; 0.4% SDS; 5 mM EDTA; 2% Triton X-100; 1 mM DTT; and Protease Inhibitor Cocktail [Roche]) and sonicated. The lysates were centrifuged at 4°C at 14,000g for 10 min and supernatants were incubated with streptavidin beads (53117; Pierce) overnight. The beads were collected and washed twice with wash buffer 1 (2% SDS in $dH_2O$) for 10 min, once with wash buffer 2 (2% deoxycholate; 1% Triton X-100; 50 mM NaCl; 50 mM Hepes, pH 7.5; and 1 mM EDTA) for 10 min, once with wash buffer 3 (0.5% NP-40; 0.5% deoxycholate; 1% Triton X-100; 500 mM NaCl; 1 mM EDTA; and 10 mM Tris, pH 8.1) for 10 min, and once with wash buffer 4 (50 mM Tris, pH 7.4 and 50 mM NaCl) for 30 min. A small portion of samples were used for Western blotting: The biotinylated proteins were eluted from the beads with 500 nM D-biotin at 98°C for 10 min. The rest of the samples were prepared for mass spectrometry analysis.

## Protein identification by mass spectrometry

To identify the biotinylated proteins by mass spectrometry, on-bead tryptic digestion was performed for each sample. Briefly, the protein-bound streptavidin beads were first washed with 50 mM $NH_4HCO_3$, then were reduced with 100 mM DTT in 50 mM $NH_4HCO_3$ at 56°C for 45 min, and alkylated with 100 mM iodoacetamide at RT in the dark for 30 min. MS Grade Trypsin Protease (Pierce) was added onto the beads for overnight digestion at 37°C (enzyme: protein ratio of 1:100). The resulting peptides were purified using C18 StageTips (Rappsilber et al, 2007). Peptides were analyzed by online C18 nanoflow reversed-phase HPLC (2D nanoLC; Eksigent) linked to a Q-Exactive Orbitrap mass spectrometer (Thermo Fisher Scientific). The data sets were searched against the human SWISS-PROT database version 2014_08. Proteome Discoverer (version 1.4; Thermo Fisher Scientific) was used to identify proteins. The final protein lists were analyzed using the STRING v10.5 database (Szklarczyk et al, 2017) and the Gene Ontology cellular component annotation (Ashburner et al, 2000) to find out already known protein interactions within the interactomes and to distribute the protein hits in terms of subcellular localization, respectively. The protein interaction network was visualized using Cytoscape 3.6.0 (Shannon et al, 2003).

## Sensitive GFP-trap pull-down for protein interaction analysis

GFP-trap pull-down experiment was performed under low detergent conditions, which was referred to as sensitive pull-down (Davies et al, 2018). Stable HeLa S3 cells expressing only GFP or wild-type CLIC4-GFP were arrested at monopolar cytokinesis. ~1 × $10^7$ cells were resuspended in 500 $\mu$l of 1× PBS containing cOmplete EDTA-free protease inhibitor cocktail (Roche) and PhosSTOP phosphatase inhibitor mixture (Roche). The cell suspensions were homogenized with a Dounce homogenizer. The cell lysates were incubated with 0.01% Triton X-100 for 20 min at 4°C and cleared by centrifugation at 4,000$g$ for 10 min 25 $\mu$l of GFP-Trap_A beads (gta-20; ChromoTek) were equilibrated with 500 $\mu$l of ice-cold dilution buffer (10 mM Tris HCl, pH 7.5; 150 mM NaCl; and 0.5 mM EDTA). Protein lysates were added onto equilibrated GFP-Trap_A beads and protein-bead mixtures were incubated for 3 h at 4°C with mild rotation. After the beads were spun down, the unbound fractions were removed. The beads were washed twice with 500 $\mu$l of ice-cold dilution buffer. Finally, GFP-Trap_A beads were resuspended in 100 $\mu$l of SDS-sample buffer and boiled at 95°C for 10 min. Input and eluate samples were analyzed by Western blotting.

## Western blotting and immunostaining

For Western blotting, the cells were lysed in 1× PBS buffer containing 0.1% Triton X-100, cOmplete EDTA-free protease inhibitor cocktail (Roche), and PhosSTOP phosphatase inhibitor mixture (Roche). Protein concentration was measured using the BCA protein assay kit (23227; Pierce). Protein samples were prepared in 2× Laemmli sample buffer containing 100 mM DTT. 7–12% SDS–PAGE gels were used for the separation of proteins. The Trans-Blot Turbo transfer system (Bio-Rad) was used to transfer proteins to nitrocellulose membranes (BA85; Whatman Protran). After blocking with 4% milk in 0.1% Tween 20 containing 1× TBS for 45 min, the membranes were incubated with primary antibodies either overnight at 4°C or 3 h at RT and incubated with secondary antibodies at RT for 1 h 30 min. The membranes were rinsed with 0.1% Tween 20 containing 1× TBS buffer for 5 min three times after antibody incubations. Proteins were visualized with the ECL Western Blotting Substrate system (32106; Pierce).

For immunostaining, cells seeded on glass coverslips were fixed in 3.2% PFA in 1× PBS for 15 min. Coverslips were washed three times with TBS-0.1% Triton X-100 buffer for 3 min. After blocking with 2% BSA in TBS-0.1% Triton X-100, coverslips were incubated with primary antibodies in 2% BSA in TBS-0.1% Triton X-100 either overnight at 4°C or 3 h at RT and incubated with secondary antibodies at RT for 1 h 30 min. The coverslips were washed three times with TBS-0.1% Triton X-100 buffer for 3 min after antibody incubations. Coverslips were embedded in Mowiol mounting medium (81381; Sigma-Aldrich).

The primary antibodies were used as 1:200 anti-CLIC4 (sc-135739; Santa Cruz), 1:200 anti-CLIC1 (sc-81873; Santa Cruz), 1:1,000 anti-$\alpha$-tubulin (2144; Cell Signaling), 1:200 anti-$\beta$-tubulin (2128; Cell Signaling), 1:5,000 anti-GFP (11814460001; Roche), 1:20,000 anti-biotin rabbit (gift from Dr Timothy J Mitchison, Harvard Medical School, Boston, MA), 1:500 anti-phospho-histone H3 (ser10) (32219; Upstate), 1:400 anti-Myc (2278; Cell Signaling), 1:500 anti-anillin (ab99352; Abcam), 1:1,000 anti-ezrin (3145; Cell Signaling), 1:1,000 anti-phospho-ERM (3726; Cell Signaling), and 1:1,000 anti-ALIX (ab88388; Abcam). 1:2,000 HRP-conjugated (Cell Signaling) or 1:1,000 Alexa Fluor 488-, 555- and 594-conjugated antimouse and antirabbit IgGs (Invitrogen) were used as secondary antibodies. DAPI (D8417; Sigma-Aldrich) and Phalloidin-iFluor 555 (ab176756; Abcam) were used to visualize DNA and actin filaments, respectively.

## PLA

Spatial associations of CLIC4 with anillin, ALIX, and ezrin were examined using Duolink PLA Kit (92101; Sigma-Aldrich) according to the manufacturer's instructions. Briefly, HeLa S3 cells were seeded onto glass coverslips and arrested at cytokinesis. The cells were fixed with 3.2% PFA for 15 min, permeabilized with 0.1% Triton X-100, treated with blocking solution for 30 min at 37°C, and then incubated with either the pair of primary antibodies (as target) or only one primary antibody (as control) in the antibody diluent solution overnight at 4°C. The cells were washed in wash buffer A for 10 min at RT and incubated with the pair of secondary antibodies conjugated to plus and minus PLA probes in the antibody diluent for 1 h at 37°C. After repeating the washing step with wash buffer A for 10 min at RT, the cells were incubated with the ligase in the ligation buffer for 30 min at 37°C. After another washing cycle with wash buffer A, the cells were incubated with the polymerase in the amplification buffer for 100 min at 37°C. Finally, the cells were washed in wash buffer B for 20 min and then with 0.01× wash buffer B for 1 min at RT. The cells were incubated with mounting medium containing DAPI for 15 min and coverslips were sealed to the slides.

## Quantification of multinucleation

To analyze the multinucleation percentage of knockout cell lines, cells were plated on glass coverslips. When the cells reached 70% confluency, they were fixed with 3.2% PFA and stained with the

anti-α-tubulin antibody and DAPI. Fields for fluorescent imaging were randomly selected for each condition. Two independent observers blind to the sample characteristics performed the image acquisition and image analyses. The number of mono- and multinucleated cells on merged images of α-tubulin and DNA was counted manually by using ImageJ software (Schneider et al, 2012). The percent ratio of multinucleated cells to total cell number was obtained for three independent experiments. The results were presented as the mean ± SEM.

### Microscopy and image analysis

The images of fixed samples were acquired using the 60× Plan Apo 1.4 NA oil-immersion objective of Nikon Eclipse 90i (EZ-C1 software) confocal microscope, 63× Plan Apo 1.4 NA oil-immersion objective of Leica DMi8 wide-field microscope or 63× Plan Apo 1.4 NA oil-immersion objective of Leica DMi8/SP8 TCS-DLS (LAS X Software) laser scanning confocal microscope. For live-cell imaging, the cells were seeded on the ibiTreat, ibidi μ-Slide 8 Well, or μ-Dish 35 mm, high plates and the image acquisition was performed using either 20× Plan Fluor 0.4 NA objective or 63× Plan Apo 1.4 NA oil-immersion objective of Leica DMi8 wide-field microscope equipped with 37°C and 5% $CO_2$ chamber. $CO_2$-independent medium (Gibco) was used during fluorescence imaging of live cells. Z-stacks were acquired every 3 min for extensive analyses of cell division. Single focal plane was used in the figures unless specified in the figure legends. Images were not deconvoluted.

To quantify the localization of endogenous or GFP-tagged CLIC4 and CLIC1 proteins at the cleavage furrow, the integrated densities were obtained by ImageJ software for the region of interests (ROIs) as shown in Figs 1C and E, 6C, and S3D. To evaluate the levels of p-ezrin and actin filaments at the cleavage furrow and polar cortex of dividing cells, the mean intensity of two ROIs at the cleavage furrow were divided to the mean intensity of two ROIs at the polar cortex as described previously (Kim et al, 2017). To quantify the maximal extension of membrane blebs, the Z frame displaying the longest membrane bleb was selected and a straight line from the cell cortex and to the tip of the bleb was drawn. The length of the line was measured by using Leica LAS X software for each cell division.

For quantification of PLA dots, 20 random fields of dividing cells were imaged in DAPI (to observe nuclei) and Texas Red (to observe red PLA dots) channels for both target and control conditions. Each channel was analyzed separately. The automated fluorescent particle analysis of each channel of every image was performed with ImageJ software as explained previously (Debaize et al, 2017). By this way, the ratio of PLA dots to the number of daughter cell nuclei was obtained for every image of each condition.

### Statistical analysis

To evaluate statistical significance, the comparison between two groups was analyzed by unpaired two-tailed $t$ test. One-way ANOVA with either Dunnett's post hoc test or Bonferroni post hoc test was used for the comparisons among multiple groups. $P < 0.05$ was considered as statistically significant. All graphs were created using GraphPad Prism 5 software.

## Supplementary Information

## Acknowledgements

The authors thank Busra Akarlar for performing the liquid chromatography-tandem mass spectrometry analyses, Ezgi Memis for the technical assistance in stable cell line generation, Dr Mohammad Haroon Qureshi for the help with microscopy, and Dr Anne Schlaitz and Aydanur Senturk for the critical reading of the manuscript. The authors gratefully acknowledge the permission to use the facilities of Cellular and Molecular Imaging Core of Koç University Research Center for Translational Medicine funded by the Republic of Turkey Ministry of Development and the Proteomics Facility of Koç University. ZC Uretmen Kagiali was funded by TUBITAK-BIDEB 2211-E Scholarship Program. N Saner was funded by TUBITAK 3501 Career Development Program (217S615). N Ozlu was funded by Science Academy (Turkey) Young Scientist Award and Installation Grant from the European Molecular Biology Organisation.

### Author Contributions

ZC Uretmen Kagiali: data curation, formal analysis, investigation, visualization, methodology, and writing—original draft, review, and editing.
N Saner: resources, data curation, funding acquisition, investigation, project administration, and writing—original draft, review, and editing.
M Akdag: methodology.
E Sanal: formal analysis.
BS Degirmenci: methodology.
G Mollaoglu: resources and methodology.
N Ozlu: conceptualization, resources, supervision, funding acquisition, methodology, project administration, and writing—original draft, review, and editing.

### Conflict of Interest Statement

The authors declare that they have no conflict of interest.

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
