## [Reviewer comments · Life Science Alliance]

CLIC4 and CLIC1 bridge plasma membrane and cortical actin network for a successful cytokinesis

Zeynep Cansu Uretmen Kagiali, Nazan Saner, Mehmet Akdag, Erdem Sanal, Beste Senem Degirmenci, Gurkan Mollaoglu, and Nurhan Ozlu
DOI: 10.26508/lisa.201900558

Corresponding author(s): Dr. Nurhan Ozlu (Koc University)

Review timeline:

Submission Date:	2019-09-19
Editorial Decision:	2019-10-15
Revision Received:	2019-12-16
Editorial Decision:	2019-12-19
Revision Received:	2019-12-20
Accepted:	2019-12-20

Scientific Editor: Andrea Leibfried

Transaction Report:

No Peer Review Process File is available with this article, as the authors have chosen not to make the review process public in this case.

October 15, 2019

Re: Life Science Alliance manuscript #LSA-2019-00558-T

Dr. Nurhan Ozlu
Koc University
Molecular Biology and Genetics
Sariyer
Istanbul 34450
Turkey

Dear Dr. Ozlu,

Thank you for submitting your manuscript entitled "CLIC4 bridges key plasma membrane and actin cytoskeleton proteins for a successful cytokinesis" to Life Science Alliance. The manuscript was assessed by expert reviewers, whose comments are appended to this letter.

As you will see, all three reviewers appreciate your data and provide constructive input on how to further strengthen your manuscript prior to publication. We would thus like to invite you to submit a revised version of your manuscript to us. The revision seems rather minor and straightforward, but please do get in touch in case you would like to discuss certain aspects further.

Thank you for this interesting contribution to Life Science Alliance. We are looking forward to receiving your revised manuscript.

Sincerely,

Andrea Leibfried, PhD
Executive Editor
Life Science Alliance
Meyerhofstr. 1
69117 Heidelberg, Germany
t +49 6221 8891 502

e a.leibfried@life-science-alliance.org
www.life-science-alliance.org

B. MANUSCRIPT ORGANIZATION AND FORMATTING:

2nd Editorial Decision

19 December 2019

December 19, 2019

RE: Life Science Alliance Manuscript #LSA-2019-00558-TR

Dr. Nurhan Ozlu
Koc University
Molecular Biology and Genetics
Sariyer
Istanbul 34450
Turkey

Dear Dr. Ozlu,

Thank you for submitting your revised manuscript entitled "CLIC4 and CLIC1 bridge plasma membrane and cortical actin network for a successful cytokinesis". I appreciate the new data and the introduced changes and think that they address well the (relatively minor) concerns that were raised by the reviewers. We would thus be happy to publish your paper in Life Science Alliance. Before sending you an official acceptance letter:

- please review the author contributions, authors listed as having contributed to "investigation" were probably also involved in manuscript preparation, which should get noted to adhere to ICMJE guidelines (<http://www.icmje.org/recommendations/browse/roles-and-responsibilities/defining-the-role-of-authors-and-contributors.html>)
 - please fill in the electronic license to publish form
- If you are planning a press release on your work, please inform us immediately to allow informing our production team and scheduling a release date.

A. FINAL FILES:

-- Summary blurb (enter in submission system): A short text summarizing in a single sentence the study (max. 200 characters including spaces). This text is used in conjunction with the titles of papers, hence should be informative and complementary to the title. It should describe the context and significance of the findings for a general readership; it should be written in the present tense and refer to the work in the third

person. Author names should not be mentioned.

B. MANUSCRIPT ORGANIZATION AND FORMATTING:

Sincerely,

Andrea Leibfried, PhD
Executive Editor
Life Science Alliance
Meyerohofstr. 1
69117 Heidelberg, Germany
t +49 6221 8891 502
e a.leibfried@life-science-alliance.org
www.life-science-alliance.org

3rd Editorial Decision

20 December 2019

December 20, 2019

RE: Life Science Alliance Manuscript #LSA-2019-00558-TRR

Dr. Nurhan Ozlu
Koc University
Molecular Biology and Genetics
Sariyer
Istanbul 34450
Turkey

Dear Dr. Ozlu,

Thank you for submitting your Research Article entitled "CLIC4 and CLIC1 bridge plasma membrane and cortical actin network for a successful cytokinesis". It is a pleasure to let you know that your manuscript is now accepted for publication in Life Science Alliance. Congratulations on this interesting work.

DISTRIBUTION OF MATERIALS:

Again, congratulations on a very nice paper. I hope you found the review process to be constructive and are pleased with how the manuscript was handled editorially. We look forward to future exciting submissions from your lab.

Sincerely,
